# Simple and Effective Derivatization of Amino Acids with 1-Fluoro-2-nitro-4-(trifluoromethyl)benzene in a Microwave Reactor for Determination of Free Amino Acids in Kombucha Beverages

**DOI:** 10.3390/ma15207365

**Published:** 2022-10-20

**Authors:** Aneta Jastrzębska, Zuzanna Gralak, Kamil Brzuzy, Anna Kmieciak, Marek P. Krzemiński, Rafał Burdziński, Marzanna Kurzawa, Edward Szłyk

**Affiliations:** 1Department of Analytical Chemistry and Applied Spectroscopy, Faculty of Chemistry, Nicolaus Copernicus University in Toruń, Gagarin 7 Str., 87-100 Toruń, Poland; 2Department of Organic Chemistry, Faculty of Chemistry, Nicolaus Copernicus University in Toruń, Gagarin 7 Str., 87-100 Toruń, Poland

**Keywords:** free amino acids, 1-fluoro-2-nitro-4-(trifluoromethyl)benzene, microwave reactor, kombucha, HPLC, NMR

## Abstract

Kombucha is a fermentation product of sweetened tea with a symbiotic culture of acetic acid and yeast bacteria, consumed worldwide for its health-promoting properties. Few reports can be found about free amino acids among the health-promoting compounds found and determined in kombucha. These compounds influence the sensory properties of kombucha, and they are precursors of bioactive compounds, which have a significant role as neurotransmitters and are involved in biological functions. The presented studies proposed a convenient, simple, and “more green” procedure of the synthesis of amino acid derivatives, assisted by microwave energy, followed by chromatographic determination. The structure of 1-Fluoro-2-nitro-4-(trifluoromethyl)benzene was used as a suitable reagent for the derivatization of free amino acids in fermented kombucha beverages prepared from selected dry fruit such as *Crataegus* L., *Morus alba* L., *Sorbus aucuparia* L., *Berberis vulgaris* L., *Rosa canina* L., and black tea. The obtained results were discussed regarding the tested beverages’ application as a source of amino acids in one’s daily diet. The obtained results point out that the proposed microwave-assisted derivatization procedure prior to HPLC analyses allows for a significant time reduction and the limitation of using organic reagents.

## 1. Introduction

Kombucha is a mildly alcoholic, slightly sweet, acidic, and effervescent beverage obtained by the fermentation of sugared tea (black, green, white, or oolong) by a symbiotic culture of bacteria and yeast (SCOBY) [1,2]. This is not novel food, but it has re-emerged since the 2000s as a popular lifestyle product with potential health benefits, produced industrially and homebrewed globally. The high demand for this beverage is due to the presence of some compounds (sugars, organic acids, ethanol, CO_2_, dietary fiber, amino acids, essential elements, vitamin C, vitamin B derivates, antibiotic substances, hydrolytic enzymes, and polyphenol) and good storage conditions [3]. According to Morales [4], many people consume kombucha drinks, presuming they are a ‘functional food’. However, these products have not been approved by institutions such as the EFSA and there is a lack of clinical trial results that confirm the effectiveness of kombucha or its ingredients on human health [4]. During the COVID-19 pandemic, the consumption of fermented beverages has increased in many countries, as they have been reported to provide many beneficiary effects [5]. In the literature [1,2,3,4], special attention was paid to the reported antioxidant, immunomodulatory, antitumoral, hypocholesterolemic, antihypertensive, and antimicrobial capacities of kombucha. Today, this beverage is prepared using different herbal raw materials as the substrate which affects the type and content of the active substances. Moreover, most of the research works have been focused on the antioxidant activity and phenolic compounds [5]. Although kombucha has been researched in detail concerning its many chemical compounds, there are not enough studies regarding the determination of amino acids (AAs). These compounds reveal a significant role in biological processes such as energy production, nutrient absorption, tissue growth, and the immune function [6]. The hydrolysis of proteins during the production of kombucha from plant material is related to the production of free amino acids (FAAs) due to the fermentation process and the quality of the raw materials. Additionally, the duration of the beverage contact with SCOBY is an essential factor for the levels of free amino acids which are produced. Jayabalan et al. [7] determined the level of amino acids in tea fungus during fermentation and observed that tea fungus is rich in these compounds.

Over the last few years, the determination of free amino acids has increased significantly for the comprehensive analysis of nutritional issues. However, there is surprisingly little which has been published about the FAA levels in kombucha beverages. These compounds create sweetness, sourness, bitterness, and an umami taste in foods [8]. Moreover, they are precursors of bioactive amines, which play a significant role as neurotransmitters and are involved in biological functions. Yet, high amounts of these compounds in one’s diet may cause adverse effects [9].

Many analytical methods for the determination of amino acids, starting with ion-exchange chromatography followed by post-column derivatization with ninhydrin and UV detection, have been reported in the literature [10]. Still, chromatographic methods such as liquid chromatography [8,11,12,13] or gas chromatography [14,15] are generally used. Non-chromatographic separation methods have also been used but less often, with the most prominent being capillary electrophoresis [6,16,17]. Despite this, high-performance liquid chromatography (HPLC) is the most implemented technique. The identification and separation of amino acids is difficult due to their high polarity, low volatility, and the absence of strong chromophoric groups. For this reason, all methods require the derivatization stage of amino acids using appropriate derivatizing reagents. The numerous derivatizing reagents with strong chromophore groups have been introduced and widely accepted. Each has specific advantages and limitations such as a long derivatization time, low stability, insufficient reproducibility, low derivative yield, and chromatographic interference caused by the reagent [18]. In the case of AAs, the derivatization reagent reacts with the amino acids’ amine or carboxyl group. Unfortunately, some amino acids have additional amines or carboxyl groups in their side chain, which can be related to multiple derivatives.

According to our previous studies [19], we proposed 1-fluoro-2-nitro-4-(trifluoromethyl)benzene (FNBT) as a derivatization reagent for the determination of free amino acids in kombucha beverages. The proposed derivatization reaction is a process of nucleophilic substitution, which is suitable for the reaction of amino acids. In the case of biogenic amines derivatization, the reaction was rapid and proceeded efficiently at room temperature. Moreover, the obtained derivatives were stable and were suitable for elution in reversed-phase HPLC [19]. However, this reagent has not been used so far for amino acid derivatization. FNBT is a close analog of the Sanger reagent (2,4-dinitrofluorobenzene, DNFB), which reacts with amino groups in amino acids to produce dinitrophenyl amino acids. Li et al. used the Sanger reagent in the precolumn derivatization of amino acids in tea infusion [20]. The reaction was carried out at 60 °C for 60 min in the dark. FNBT and DNFB belongs to halonitrobenzenes, a class of typical derivatization reagents that can specifically react with amino groups.

The microwave technique has recently been used in derivatization for its considerably short treatment time and low energy consumption. It has become a viable replacement to conventional heating [21]. According to Sajid and Płotka-Wasylka [22], the application of microwaves, which perform the derivatization process under mild conditions and often accelerate the chemical conversion of analytes, is also in accordance with green analytical chemistry principles. In addition, a strict control over the temperature and the time of irradiation allows for focusing on a small sample volume, resulting in an increased precision. This method was developed for the derivatization of AAs with DNFB before capillary electrophoresis detection [23]. In this case, the process was not performed with a dedicated microwave reactor but with a modified household microwave oven. Nevertheless, the studies in the literature have indicated that the application of microwaves for amino acid derivatization is still rare.

The goal of this paper was the chromatographic determination of free amino acids in kombucha based on different dry fruits such as hawthorn (*Crataegus* L.), white mulberry (*Morus alba* L.), red rowan (*Sorbus aucuparia* L.), barberry (*Berberis vulgaris* L.), dog rose (*Rosa canina* L.), and black tea. According to our studies, there is no information about the FAAs content in such beverages in the available literature. For this purpose, we proposed a modified procedure of amino acid derivatization with FNBT using a microwave-assisted method prior to HPLC. The FNBT derivatives of twenty-one amino acids were studied under simple conditions in an aqueous medium. The obtained products were tested on their purity, while the structures were confirmed by ^1^H, ^13^C, and ^19^F NMR spectra. Finally, the reverse phase HPLC with UV-Vis detection was applied to determine the obtained derivatives and food analysis. The achieved results were discussed regarding the application of tested beverages as a source of amino acids.

## 2. Experimental Section

### 2.1. Reagents and Instruments

The analytical grade: arginine hydrochloride (Arg), aspartic acid (Asp), asparagine (Asn), β-alanine (Ala), cysteine (Cys), glutamine (Gln), glutamic acid (Glu), glycine (Gly), histidine (His), isoleucine (Ile), leucine (Leu), lysine monohydrochloride (Lys), methionine (Met), ornithine (Ort), phenylalanine (Phe), proline (Pro), serine (Ser), threonine (Thr), tryptophan (Trp), tyrosine (Tyr), valine (Val), 1-fluoro-2-nitro-4-(trifluoromethyl)benzene (FNBT), ethyl acetate, acetonitrile (ACN, HPLC grade), methanol (HPLC grade), tetrahydrofuran (THF, HPLC grade), and deuterium oxide were purchased from Sigma Aldrich (Poznań, Poland). Hydrochloric acid, boric acid, sodium tetraborate, sodium carbonate anhydrous, and sodium bicarbonate were purchased from Alchem Grupa Sp. z o.o. (Toruń, Poland).

The microwave derivatization was performed with the microwave synthesizer Discover 2.0 (CEM Corporation, Matthews, NC, USA). The HPLC system (Shimadzu Corp, Kyoto, Japan), equipped with an autosampler SIL-20AC HT and a photodiode multiwavelength detector (SPD-M20A Prominence Diode Array Detector, Shimadzu Corp, Kyoto, Japan) was applied. Analyses were carried out on an EC 250/4 Nucleoshell RP 18.5 µm (Macherey-Nagel, Macherey-Nagel GmbH&Co. KG, Düren, Germany) at 35 °C. The chromatographic data were recorded and processed by the LC solution program version 1.23 SP. NMR spectra were recorded on a Bruker Avance III 400 MHz spectrometer (Bruker, Ettlingen, Germany) at 400 MHz for ^1^H, 100 MHz for ^13^C, and 375 MHz for ^19^F frequency resonances at 298 ± 1 K using deuterium oxide as a solvent. The melting points were obtained in open capillary tubes using electrothermal digital melting point apparatus from Cole-Parmer Ltd., Staffordshire, UK, and are uncorrected. The specific rotation was measured with a Bellingham and Stanley ADP400 series polarimeter (Bellinghent Stanley Ldb. Tunbridge Wells, UK).

### 2.2. Preparation of Kombucha

The dry fruits of hawthorn (*Crataegus* L.), white mulberry (*Morus alba* L.), red rowan (*Sorbus aucuparia* L.), barberry (*Berberis vulgaris* L.), dog rose (*Rosa canina* L.), and black tea were used. About 5 g of the plant material was weighed on an analytical balance, crushed in a mortar, and quantitatively transferred to a beaker filled with 200 mL of distilled boiling water, stirred, and left to chill. The mixtures were transferred into a glass bottle (2 L), and 10 mL of apple cider vinegar and 100 g of sucrose were added. Afterward to the mixture, 800 mL of distilled water was added, and SCOBY was introduced. The fermentation process was carried out for 14 days at a temperature of 20–22 °C. After completion, the SCOBY was removed and the kombucha was transferred into the storage bottle for further analysis.

### 2.3. Microwave Assisted Derivatization Process—General Procedure

In order to obtain the amino acid derivatives with FNBT, alkaline solution and 10% excess of FNBT were applied (Figure 1).

**Scheme 1 materials-15-07365-sch001:**
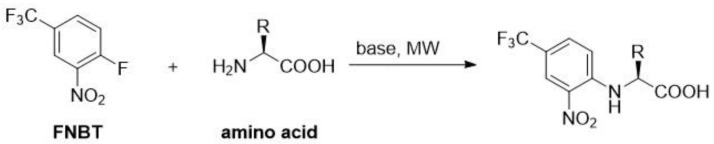
The reaction of amino acid (1.5 mmol) with FNBT (1.65 mmol) was carried out in carbonate buffer (pH = 10; 5 mL) at 130 °C (20 min) employing microwave (MW) irradiation. In the case of amino acids with two amino groups (lysine and ornithine), an appropriate excess of FNBT (2.2 eq, 3.3 mmol) was used. The obtained derivatives are presented in Figure 1.

**Figure 1 materials-15-07365-f001:**
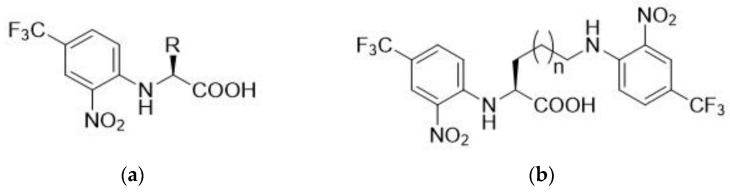
Chemical structures of amino acids derivatives with FNBT; (**a**) (2-Nitro-4-(trifluoromethyl)phenyl)-*L*-amino acid; (**b**) N^2^,N^6(5)^-Bis(2-nitro-4-(trifluoromethyl)phenyl)-*L*-amino acid.

After the completion of the reactions, the solvent was removed on a rotary evaporator under a reduced pressure, and the product was purified by column chromatography on a silica gel. The structures of amino acids derivatives were confirmed by ^1^H, ^13^C, and ^19^F NMR spectra and characterized by optical properties and melting point temperatures. Finally, the obtained products were dissolved in methanol (HPLC grade, 10 mL), and the RP-HPLC-DAD technique was applied for analysis.

In the case of kombucha beverages, derivatization reactions were carried out in the same way: 10 mL of the samples were mixed with carbonate buffer until the pH = 10, and FNBT (300 µL) was added. The reactions were carried out at 130 °C for 20 min with MW irradiation. Next, they were twice extracted with 10 mL of ethyl acetate, and the organic layer was collected. The combined extracts were dried over anhydrous MgSO_4_ and evaporated to give a crude product. Finally, the obtained products were dissolved in methanol (HPLC grade, 5 mL) and analyzed by RP-HPLC. Additionally, the lack of impact of the organic reagents on the synthesis of the derivatives was confirmed by the reaction performed with acetonitrile (1 mL).

### 2.4. Chromatographic Analysis

The obtained derivatives of amino acids were determined using RP-HPLC, as reported by Li et al. [20]. The mobile phase was: 5 mM sodium acetate/tetrahydrofuran 95:5, *v*/*v* (solvent A) and methanol/water 80:20, *v*/*v* (solvent B), and the gradient conditions were: 0 min A: 90%; 20 min A: 80%; 25 min A: 50%; 35 min A: 45%; 50 min A: 0%; and 60 min A: 90%. The total flow rate was 1 mL/min; the temperature was 35 °C; the injection volume was 20 µL; and the detection wavelength was 254 nm. The chromatographic data were recorded and processed by the LC solution program, version 1.23 SP (Shimadzu Corp, Kyoto, Japan).

### 2.5. Analytical Performance

Quantification was performed using the calibration curves of the respective AA standards. The calibration curves were constructed using the standard solutions of analytes in the different concentration ranges (Table 1). The linearity, limits of detection, quantitation, precision, and accuracy of the proposed method were verified. The amino acid derivatives in the kombucha beverages samples were identified according to the retention times of the corresponding standards. Five independent samples were used and analyzed in triplicate for each of the tested beverages. A one-way ANOVA was performed to determine the significant differences between the data (*p* < 0.05).

### 2.6. ^1^H, ^13^C, ^19^F NMR Studies

The ^1^H, ^13^C, and ^19^F NMR spectra for most amino acid derivatives from FNBT were recorded in deuterated water. In the case of derivatives of several amino acids (e.g., methionine), due to its low solubility in deuterium oxide, methanol-d_4_ was used as the spectral recording solvent.

## 3. Results and Discussion

### 3.1. Chemistry

The reaction of amino acids with FNBT follows the nucleophilic substitution mechanism on the aromatic ring. In the case of FNBT, the fluorine atom undergoes nucleophilic substitution with a free amino group derived from an amino acid. In the ongoing reaction, it is essential to shift the acid-base equilibrium of the amino acid so that the amino group is in a non-protonated form, which allows the nucleophilic substitution reaction to take place.

The microwave-assisted derivatization (MAD) seems to be an alternative to the conventional heating of the samples during the derivatization process due to several advantages: a shortening of the reaction time, reduction in energy consumption, simplicity, efficiency, elimination of side reactions, higher yield under milder reaction conditions, higher purity of the products formed, and suppression in the rate of by-product formation [24]. In this paper, we proposed a microwave-assisted amino acid derivatization with FNBT without the application of other organic reagents. In order to obtain the best derivatization yield, the reactions’ pH and the acetonitrile presence were tested. For this purpose, the solutions of the selected amino acids (tryptophan, phenylalanine, lysine, and histidine) were mixed with the appropriate amount of FNBT in the presence of borate (pH = 9; 5 mL) and carbonate (pH = 10; 5 mL) buffers, and the conversions for these reactions were determined by analyses of the ^1^H NMR spectra (Appendix A).

The obtained results indicated a pH value that = 10 (carbonate buffer) was the better choice, whereas the presence of an organic solvent did not affect the synthesis of the AA derivatives. It should be added that FNBT is insoluble in water. However, applying microwave energy allowed the synthesis of the derivatives in the water environment (aqueous solution) so that the final procedure could be simplified. The yield of the reaction (Figure 2) varied from 96.54% (Asn) to 100% (Trp and Phe).

### 3.2. Structural Characteristics of the Obtained Derivatives

For all amino acids with one free primary amino group in their structure, based on the analysis of the number of signals and the integration of individual peak areas in the ^1^H NMR spectra, the substitution of one group derived from the derivatizing reagent (FNBT) was unambiguously confirmed. On the other hand, in the case of basic amino acids with two primary amino groups (lysine and ornithine), the number of signals and their integration in the ^1^H NMR spectra confirmed the substitution of two aromatic groups. In the case of amino acids with NH_2_ groups of a non-amino nature (amide groups in glutamine and asparagine, guanidine system in arginine), only the substitution of amino groups in AAs was noted, which confirms the mechanism of nucleophilic substitution in the aromatic ring. Moreover, in the case of lysine and ornithine, two signals from the CF_3_ groups with the same integration value are observed in the ^19^F NMR spectra. The full description of ^1^H, ^13^C, and ^19^F NMR of amino acid derivatives are presented in the Appendix A.

### 3.3. The Chromatographic Separation of the Amino Acids Derivatives

The results of the linear calibration ranges, coefficient of determination, DL and QL, and the retention time for amino acids standard solutions using the HPLC procedure are listed in Table 1.

In the proposed linear range, the response value was linearly regressed to the amino acid concentration, and the linear correlation coefficient varied from 0.9954 (Asn) to 0.9999 (Gly, Thr). The coefficient of variation (CV) for the retention time is below 0.55% for all tested amino acids, which indicates a good intraday repeatability. The calculated QLs were sufficiently low for the determination of the amino acids (except Lys and Asn), whereas the DLs were below the lowest concentration.

### 3.4. Determination of FAAs in Kombucha Beverages

The developed procedure was evaluated by an analysis of the free amino acids in the kombucha beverages. Based on preliminary research, the derivatization process was carried out in an aquatic environment and for comparison with 1 mL of acetonitrile. The obtained results for the two samples: kombucha from hawthorn and black tea kombucha, are listed in Table 2.

The obtained results for the derivatization procedure in the water environment and the presence of ACN are comparable (a one-way ANOVA and Duncan test). Statistically significant differences for the determined individual FAAs in black tea kombucha after two derivatization procedures were observed only for three amino acids (Pro, Ile, and Ort), whereas for Arg in the hawthorn beverage. This suggested that the usage of MV energy enabled the synthesis of the derivatives, despite the poor solubility of FNBT in aqueous solutions.

The developed procedure was applied to determine the free amino acids in the remaining beverages. The obtained results are listed in Table 3.

Many studies on the relationship between diet and health increased the interest in proposing new functional foods rich in nutrients. A plant-based edible such as kombucha is recognized as potentially pro-health. Several factors impact the concentration of kombucha constituents, such as the composition and concentration of raw materials, sugar concentration, fermentation time, and temperature. The typical production of kombucha beverages is based on black, green, or oolong tea [1]. In this study, we proposed hawthorn, white mulberry, red rowan, barberry, and dog rose as raw materials for kombucha production and obtained results that were compared with black tea. It should be noted that there is very little data on the content of free amino acids in these plants; moreover, the fermentation process, temperature, and time influence the amount of the discussed compounds.

All analyzed beverages contained a significant number of amino acids. The total amino acid content was the largest in the hawthorn kombucha, whereas the lowest was in the barberry kombucha. The hawthorn is a valuable source of medicinal raw material widely used for cardiovascular disorders, which indicates that the obtained product of fermentation of this raw material may reveal better health properties [25]. Moreover, the tested product can supplement our diet with valuable ingredients, such as free amino acids.

The lower level of total FAAs in black tea kombucha (still, significant differences between the results were calculated) was observed for white mulberry, followed by dog rose and red rowan beverages. The mulberry is popular in European countries due to its nutritiousness, deliciousness, safety, and abundant active benefits. The mulberry fruit has indicated potent antioxidant, anti-inflammatory, hypolipidemic, neuroprotective, and antitumor activities due to phenolic constituents (flavonoids, anthocyanins, and phenylpropanoids) [26]. In the case of dog rose and red rowan, both plants fruits have been used in folk medicine for a long time. They are a good source of vitamins, polysaccharides, organic acids, dietary polyphenols, and minerals. Moreover, the red fruits of dog rose are rich in carotenoids, tannins, pectin, amino acids, and fatty acids [27]. Enriching kombucha with ingredients derived from these fruits may be vital for our health.

Usually, nine protein amino acids (isoleucine, leucine, valine, lysine, methionine, phenylalanine, threonine, tryptophan, and histidine) are considered indispensable. Lysine and threonine are considered as strictly indispensable, whereas among the other amino acids, arginine, cysteine, proline, tyrosine, glutamine, and glycine are classified as conditionally indispensable, due to specific physiological and pathophysiological situations. The highest and lowest amount of essential amino acids were determined in the same beverages for the total content of FAAs. Interestingly, black tea kombucha was the second in content for essential FAAs. Their content was similar to hawthorn and red rowan kombucha. It should be added that the calculated ratio of total essential amino acids to the total content of FAAs varied from 66% (white mulberry kombucha) to 71% (barberry kombucha), which indicated that a high nutritional value characterized all tested beverages. They can be a valuable essential amino acids diet supplement.

In the case of individual free amino acids, their contents varied between the tested kombuchas. Among the tested FAAs, Asp, Glu, and Thr were not detected, whereas Val was below the detection limit. Asp and Glu are non-essential amino acids under normal conditions, whereas Thr and Val must be provided in the diet. According to Jayabalan et al. [7], tea fungus is rich in these compounds, and lysine was the dominant essential amino acid, followed by isoleucine and leucine. In contrast, in non-essential amino acids, glutamic acid, alanine, aspartic acid, and proline were present in higher concentrations. However, in our study, the main amino acid in the four beverages was tryptophan, except in red rowan and dog rose kombucha. Moreover, a higher Trp content than tea kombucha was observed in hawthorn and white mulberry beverages. This amino acid is essential for the proper functioning and maintenance of health, and after consumption, it is transformed into bioactive metabolites, such as serotonin, melatonin, and kynurenine. According to Friedman [28], dietary tryptophan and its metabolites seem to have the potential to contribute to the therapy of many diseases (depression, autism, cognitive function, or cardiovascular disease). Since Trp cannot be synthesized de novo and must be supplied through the diet, its presence in tested beverages can be seen as another positive aspect of the fermentation process. Moreover, dietary tryptophan is used for protein synthesis and metabolized via the serotonin and kynurenine pathways [9].

Among other essential FAAs, Ile, Phe, and Lys were at a significant level, but their content depended on the type of raw material. Ile was the second dominant amino acid determined in hawthorn and black tea kombucha and the third in white mulberry, red rowan, and dog rose beverages. This amino acid is one of the branched-chain amino acids. It is critical for the physiological functions of the whole body. Moreover, it can improve the immune system, including immune organs, cells, and reactive substances. On the other hand, the results presented by Yu et al. [29] reveal isoleucine as a key regulator of metabolic health and suggest reducing dietary isoleucine as a new approach to treating and preventing obesity and diabetes. Apart from Ile, black tea kombucha was a rich Lys source, showing a similarity with previous studies about tea fungus [7]. Alternatively, Yılmaz et al. [9] determined free amino acids in a different tea and found Asp and Glu to be higher when compared to other amino acids. Even so, in tested black tea kombucha, these amino acids were not detected, probably because we did not test the tea leaves but the obtained beverages.

Methionine was detected only in two samples: red rowan and barberry beverages at a low level. It is an essential amino acid and considered to be one of the most harmful when consumed in excess.

Histidine is classified as a nutritionally essential amino acid and must be obtained from the diet. This AA is considered indispensable in healthy adults [30]. Higher dietary histidine is associated with improved glucoregulatory outcomes. At low doses, histidine may benefit the cognitive function and influence eating behavior. Based on the obtained results (Table 3), a significant level of this amino acid compared to black tea kombucha was observed for hawthorn and red rowan beverages.

Arginine was a conditionally essential amino acid, and it was determined in all samples, whereas its highest value was observed for white mulberry and barberry beverages. It should be added that only dog rose kombucha was characterized by a lower value of Arg compared to black tea. This amino acid, especially in the free form, has insulin-promoting and secretory effects, promoting growth and wound healing. Arg may confer metabolic, vascular, and immunological benefits for an individual in a state of stress [31]. In this context, proposed beverages seem to be a better source of Arg than typical kombucha.

Apart from the vital role for the human body, free amino acids are precursors for biogenic amines; moreover, fermentation increases the formation of these compounds. Among FAAs determined in our study, tryptophan, lysine, phenylalanine, histidine, tyrosine, and ornithine are precursors for tryptamine, putrescine, phenylethylamine, histamine, tyramine, and putrescine, respectively. These compounds have potential toxicological and health implications on the human body. Yet, the number of articles and reviews that discuss the concentration of Bas in kombucha is still limited [32].

It should be added that kombucha can be prepared using different raw materials as the substrate. This beverage has many desired and confirmed effects on human health. However, the antioxidant, antimicrobial properties, or probiotics effect was usually discussed in addition to the physicochemical parameters. Additionally, the levels of organic acids, vitamins, ethanol, lipids, proteins, polyphenols, and minerals were described. Surprisingly, the number of studies related to the evaluation of the content of free amino acids is still limited. Due to the constant popularity growth of kombucha and the ease of obtaining it, both in controlled and home conditions, knowledge of the content of various components that affect our health, including free amino acids, is increasingly important.

## 4. Conclusions

A simple and sensitive method of free amino acids determination using HPLC coupled with FNBT microwave-assisted derivatization was proposed and applied to kombucha beverage samples. The presented approach characterized a statistically acceptable precision, accuracy, and sensitivity, and a significant reduction in the sample preparation time, due to the use of microwave support, was achieved as well. A considerable advantage of the microwave technique in synthesizing derivatives comes from eliminating the organic reagents (environmentally friendly procedure), increased process efficiency, and overall analysis time reduction.

The results indicated that using fruits as alternative substrates for kombucha beverages was favorable due to the increased free amino acids levels. The study confirmed that most tested beverages revealed a better or similar FAAs content than black tea kombucha. Among the tested beverages, special attention should be paid to hawthorn kombucha as a good source of free amino acids.

## Data Availability

Not applicable.

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
