# Peer review of "Simple and Effective Derivatization of Amino Acids with 1-Fluoro-2-nitro-4-(trifluoromethyl)benzene in a Microwave Reactor for Determination of Free Amino Acids in Kombucha Beverages"

_materials, 2022, doi:10.3390/ma15207365_

Round 1
Reviewer 1 Report
These advises could be taken in consider for improving manuscript:
Page 3, Line 98: Please revise as ‘According to Sajid and Płotka-Wasylka [22]’
Page 4, Line 171: please delete ‘’
Figure 2 is should be revised
For Table 3, please add statistical evaluation for each amino acid
What is the main question addressed by the research? Is it relevant and interesting?
The main question of the research is determination of the free amino acid content black tea kombucha samples produced with dry hawthorn, white mulberry, red rowan, barberry, dog rose. The authors utilized their method with FNBT (1-fluoro-2-nitro-4-(trifluorome- 83 thyl)benzene) derivatization for free amino acid determination and assisted by microwave. They also, used this method for the very common fermented product: wine (Jastrzębska, A.; Piasta, A.; Kowalska, S.; Krzemiński, M.; Szłyk, E. A new derivatization reagent for determination of biogenic 465 amines in wines. J. Food Compost. Anal. 2016, 48, 111–119. doi.org/10.1016/j.jfca.2016.02.012.). This shows they are experienced at the interpretation of the analyses in fermented products. I find preferring kombucha as a research material and evaluating its potential in this concern valuable.
How original is the topic? What does it add to the subject area compared with other published material?
Kombucha is a remarkable fermented product for food science. Interest and awareness of this product is increasing recently. As a probiotic fermented product, it has a good effect on immunity. Fermented products are important for human consumption for healthier nutrition. For this reason, I personally find fermented products important. The increase in the number of studies on kombucha in recent years confirms the idea as well. But, as mentioned in the manuscript, most of the research has been focused on antioxidant activity and phenolic compounds. Also, the effect of the usage of different materials. Determination of free amino acid content of kombucha is the unique part of the study. Authors also indicated the importance of the free amino acid content of the foods in the introduction part well.
Is the paper well written?
Yes, the paper is well written, because they addressed their goal well and structured the manuscript in this concern. Importance of free amino acid content, the gap in the literature about kombucha in terms of free amino acid content, why they prefer the methodology, outstanding sides of the method, and chemical conversion in the method are well explained.
Is the text clear and easy to read?
Expressions and construction flow of the manuscript are easy to read. There is no disconnection between sections and subsections.
Authors have writing styles that is similar, and the different at the same time from classical manuscript styles. This makes it easy to comprehend the subject. For this reason, I recommended the minor revision. English usage could be improved for classical style.
Are the conclusions consistent with the evidence and arguments presented?
Result and discussion parts came up well and the conclusions consistent with the goals of the manuscript. Also, the conclusion is mentioned simply and clearly.
Do they address the main question posed?
Yes, they addressed the main question posed well and in a detailed way.
I recommended acceptance of the manuscript. Because figure 2 should be revised and minor failures should be corrected. Also, deficiencies of manuscript could be classified as editorial and easy to eliminate. I believe that the manuscript has great potential for future kombucha studies by means of free amino acid determination.
Author Response
DETAILED RESPONSES TO REVIEWER 1
We want to take the time to thank you for all your efforts in increasing readability and avoiding ambiguities within the presented study. We carefully checked every point suggested, and now the manuscript in a revised form.
- Page 3, Line 98: Please revise as ‘According to Sajid and Płotka-Wasylka [22]’
- Page 4, Line 171: please delete ‘’
REPLY: These changes have been made as suggested by Reviewer.
- Figure 2 is should be revised
REPLY: The Figure 2 was changed and improved for better clarity.
- For Table 3, please add statistical evaluation for each amino acid
REPLY: As suggested by Reviewer statistical evaluation for each amino acid was added (one-way ANOVA and Duncan test, p<0.05). Additionally, standard deviations for the sum of all free amino acids and the sum of essential free amino acids including arginine have been completed.
- Authors have writing styles that is similar, and the different at the same time from classical manuscript styles. This makes it easy to comprehend the subject. For this reason, I recommended a minor revision. English usage could be improved for the classical style.
REPLY: We have revised our paper carefully in respect of English. We believe that the language is now acceptable.

Reviewer 2 Report
Here I am sending you my comments on submitted article entitled “Simple and effective derivatization of amino acids with 1-fluoro-2-nitro-4-(trifluoromethyl)benzene in a microwave reactor for determination of free amino acids in kombucha beverage” to Materials Journal.
The article shows an interesting method of free amino acids determination using HPLC coupled with FNBT microwave-assisted derivatization (on kombucha beverages samples) in order to have a significant reduction of sample preparation time.
However, there are some comments that I want to share.
MINOR COMMENTS:
Page 2 line 45: please add some references
Page 2 line 60-80: please improve the English language. The word “however” is repeated six times.
Author Response
DETAILED RESPONSES TO REVIEWER 2
We would like to take the time to thank you for putting your efforts into increasing readability and to avoid ambiguities within the presented study. We carefully checked every point suggested and present the manuscript now in a revised form.
We want to take the time to thank you for all your efforts in increasing readability and avoiding ambiguities within the presented study. We carefully checked every point suggested, and now the manuscript in a revised form.
Page 2 line 45: please add some references
REPLY. The references were added as suggested by Reviewer.
Page 2 line 60-80: please improve the English language. The word “however” is repeated six times.
REPLY: We have revised our paper carefully in respect of English and all changes in the text are marked up using the “Track Changes” function. We believe that the language is now acceptable.
